# ATAT: Automated Tissue Alignment and Traversal

**Steven Song**
Department of Computer Science
University of Chicago
songs1@uchicago.edu

**Emaan Mohsin**
Department of Pathology
University of Chicago
emohsin@uchicago.edu

**Andrey Kuznetsov**
Department of Pathology
University of Chicago
akuznets@uchicago.edu

**Christopher Weber**
Department of Pathology
University of Chicago
cweber1@uchicago.edu

**Robert Grossman**
Department of Computer Science
University of Chicago
rgrossman1@uchicago.edu

**Aly Khan**
Department of Pathology
University of Chicago
aakhan@uchicago.edu

## Abstract

The spatial geometry of tissue biopsies reveals complex landscapes of cellular interactions. With the advent of spatial transcriptomics (ST), the ability to measure RNA across these intricate terrains has significantly advanced. However, without a pathologist's insight to delineate regions of interest, modeling gene expression transitions across specific regions becomes a daunting task. A case in point is grading the severity of inflammatory bowel disease (IBD) across the intestinal wall while identifying the organization of immune cell types across the tissue layers; such characterization will be essential in the push for precision medicine. Yet the challenge to harness ST data to decipher spatially dependent transcriptional programs in a scalable and automated manner remains a well acknowledged barrier to wider implementation. Our study aims to: (1) Utilize hematoxylin and eosin (H&E) stained images for automated segmentation of histological regions and (2) Simulate the gene expression transition across these histological layers within a single algorithmic framework. To these ends, we present ATAT: Automated Tissue Alignment and Traversal. With our approach, we automate the integration of H&E stained images with spatial transcriptomics and simplify the investigation of important biomedical questions, such as characterization of inflammatory conditions across intestinal walls.

## 1 Introduction

Spatial transcriptomics (ST) has become an increasingly utilized and powerful tool for analyzing the spatial geometry of tissues, facilitating the discovery of novel cellular interactions and disease biology [1]. Analysis of ST data often requires manual annotation and alignment of these data by pathologists, a labor intensive and expensive task that can be prohibitive for the wider adoption of ST. There is a clear need for an automated approach to tie together spatial data with digital histology, thus enabling scientific discovery [2].

Spatial geometry of cellular interactions in tissue reveals a complex landscape essential for understanding disease. The advent of ST has facilitated the measurement of RNA across these complex landscapes. Here the measurement of genes overlaid on a histological image provides a multi-dimensional view of tissue. These data allow us to better understand how cells are organized relative to different histological regions. Such a characterization is critical for understanding drug response and disease diagnosis. For example, the severity of inflammatory conditions grows as immune cells infiltrate the intestinal mucosa and the depth of infiltration varies by disease etiology [3]. The growing recognition of the relative position of immune cells across physiologically and histologically defined

NeurIPS 2023 AI for Science Workshop.

compartments points to the powerful opportunity for leveraging ST data for a deeper understanding of immune interactions and precision health. However, with the growing amount of ST data that is collected, the lack of pathologist-delineated regions of interest makes modeling gene expression across specific areas challenging [4, 5].

Scalable solutions for modeling transcriptional dynamics along a tissue axis have not received significant attention. This may be because many methods rely solely on the ST data without incorporating the H&E image, losing critical information in tissue morphology [6, 7, 8, 9, 10, 11, 12]. While the tissue morphology is likely reflected in the transcriptional phenotype, ST data is often sparse [13] and thus it can be difficult to accurately predict the tissue transitions based on ST data alone. Additionally, methods which do achieve reasonable performance in aligning tissue layers require manual segmentation of tissue samples by experts, a time-consuming and expensive task, or are unable to parse tissue samples with complex non-layered morphologies [6]. Furthermore, foundational models which achieve excellent performance in automated tissue segmentation are prone to often imperceptible batch effects, requiring additional QA when utilizing these models [14].

**Our contribution**    In this paper, we introduce ATAT: Automated Tissue Alignment and Traversal, an algorithm which, to our knowledge, is the first algorithm that utilizes self-supervised contrastive learning over the H&E image to align and traverse ST data. We propose an integrated algorithmic framework that is tailored to the dataset and utilizes the H&E images of a tissue biopsy to automatically chart a path across a sample with any morphology. This enables us to model transcriptional changes across histological layers, thereby reducing the burden of manual segmentation of tissue samples. We aim to provide a scalable and efficient solution for utilizing ST data in scientific discovery.

## 2 Methods

### 2.1 Data collection and preprocessing

We collected tissue samples from 12 colon and 4 stomach resections under protocol IRB21-0929 approved by the University of Chicago Institutional Review Board. Of the 12 colon sections, 4 are from patients with ulcerative colitis (UC), 4 are from patients with Crohn's disease (CD), 3 are from patients with *Clostridoides difficile* (C. diff) infection, and 1 is from a patient with normal colon. Of the 4 stomach sections, 3 are from patients with *Helicobacter pylori* (H. pylori) infection and 1 is from a patient with normal stomach. Each tissue sample was processed using the 10x Genomics Visium Spatial Gene Expression platform. Sequencing data was preprocessed using the Space Ranger software v1.3.1. For each H&E stained slide, the ST data is resolved as a hexagonal grid of spots. Within the square 6.5mm tissue capture area, there are 78 rows and 64 columns of spots, with each spot covering an area 55 μm in diameter, though the exact number of spots depends on the size of the tissue sample.

### 2.2 Gene filtering and normalization

We filter and normalize our ST data per slide. Because these data are often sparse, we begin by only selecting genes which have 15% or more of the total spots with non-zero UMI count, as in [6]. We then normalize our gene expression matrix, $X$, within each spot (i.e. tile), similar to normalization of single-cell RNA seq counts:

$$\tilde{X}_{s,g} = \log\left(1 + tX_{s,g}\left(\frac{\sum X}{|S|}\right)^{-1}\right)$$

where $\tilde{X}_{s,g}$ is the normalized count of gene $g$ at spot $s$, $t$ is the target average count per spot (we set $t = 10^4$), $S$ is the set of all spots, and $\frac{\sum X}{|S|}$ represents the average count per spot.

### 2.3 Automated tissue alignment and traversal

To plot ST data across tissue layers, we first learn a spatially aware representation of each image tile centered on the ST spots and calculate a similarity score between each pair of adjacent tiles. We then find the shortest path between a user selected start and end point on the slide, using the similarity

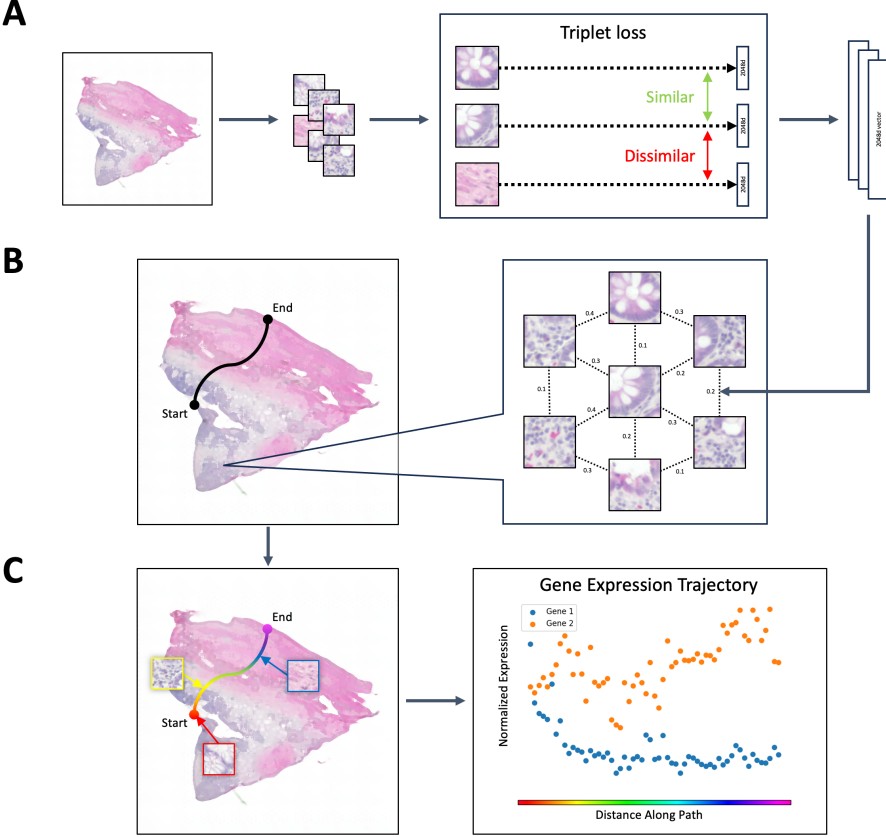

Figure 1: An overview of the Automated Tissue Alignment and Traversal (ATAT) algorithm. (A) Learning spatially aware tile representations. A tissue sample H&E slide is cut into tiles and used to train a convolutional neural network using triplet loss to learn visually similar tiles and gradual transitions between adjacent tiles. (B) Path traversal through the lattice structured graph. A similarity score between adjacent tiles is calculated using the learned tile representations. A path is traversed between a user selected start and end point using the similarity score as edge weights for a shortest path algorithm. (C) Aligning tiles to the traversed path. The tiles from the wider slide are aligned to the path using the similarity score between the learned tile representations. The gene expression at each point along the path is averaged across the set of tiles which are most visually similar to the path tile.

---

**Algorithm 1** ATAT Algorithm

1: learn spatially aware embedding function $f$ over tile set $S$
2: set start tile $a \in S$ and end tile $b \in S$ from user input
3: construct tile graph $G$ weighted by similarity matrix $W$
4: find shortest path $P^*$ between $a$ and $b$ through $G$
5: compute path clusters $S_p$ for each tile $p \in P^*$
6: average gene expression $\hat{X}_p$ along path $P^*$

---

score as edge weights. Finally, we cluster each tile on the slide to the tiles along the path and use these cluster assignments to average the gene expression along the path. Fig 1 depicts a graphical overview of ATAT and the pseudocode for ATAT is described in Algorithm 1.

### 2.3.1 Learning spatially aware tile representations

We first generate tiles from H&E slides. Each tile is a square with side length equal to the spot diameter in pixels and centered on the spots in the hexagonal grid. We then normalize each tile by subtracting the dataset mean and dividing by the standard deviation for each channel.

We train a convolutional neural network (CNN) using triplet loss over the dataset to learn a spatially aware tile representation (Fig 1A). For the CNN, we utilize a resnet50 architecture with a 3 layer multilayer perceptron on top with hidden dimension 2048, as in [15]. For our experiments, we train the model for 1000 epochs with batch size 768, learning rate 0.05, and SGD optimizer with momentum 0.9 and weight decay 1e-4.

Triplet loss is a margin-based loss function used to learn useful embeddings by distance metric learning. We employ triplet loss to cluster tissue tiles into distinct histological regions, ensuring that tiles of similar histological types are pulled closer while those of different types are pushed apart in the embedding space. Formally, given a triplet of tiles $(a, p, n)$ where $a$ is an anchor tile, $p$ is a positive tile randomly sampled from the tiles immediately adjacent to $a$, and $n$ is a negative tile randomly sampled from all other tiles in the dataset, the triplet loss is defined as:

$$L_{\text{triplet}} = \max(0, \ d(f(a), f(p)) - d(f(a), f(n)) + \alpha)$$

where $d$ is a distance metric (we use mean squared error for training), $f$ denotes the embedding function learned by the CNN, and $\alpha$ is the margin that is enforced between positive and negative pairs (0.001).

### 2.3.2 Path traversal through the lattice structured graph

After triplet loss training, we then leverage the hexagonal lattice structure of the spatial profiling array to define a graph $G(V, E)$, where $V$ represents the vertices (tissue tiles) and $E$ represents the edges (adjacent tile relationships). We define a weighted adjacency matrix $W$ based on the tile similarity inferred from triplet loss:

$$W_{i,j} = d(f(s_i), f(s_j))$$

where $d$ is a distance metric (we use Euclidean distance for the adjacency matrix), and $s_i, s_j$ are tissue tiles.

After defining a start vertex $a$ and end vertex $b$ from user input, we find the shortest path from $a$ to $b$ through this weighted graph that traverses the histological landscape (Fig 1B); Dijkstra's algorithm or a similar shortest-path algorithm can be employed to navigate through the lattice structure:

$$P^* = \operatorname*{argmin}_{P \in P_{a \to b}} \sum_{i=0}^{|P|-2} W_{i,i+1}$$

where $P_{a \to b}$ denotes the set of all possible paths through the grid from $a$ to $b$, and $P^*$ is the optimal path traversing through the tissue landscape.

### 2.3.3 Aligning tiles to the traversed path

Upon determining the optimal path $P^*$ through the lattice-structured spatial profiling array, we aim to align the rest of the tiles in the slide to the tiles along the path. This is achieved by assigning membership of any tile to a tile in the path, where each tile along the path acts as a centroid for a cluster. The length of the path signifies the number of clusters, $|P^*| = k$. Formally:

$$C(s) = \operatorname*{argmin}_{p \in P^*} d(f(s), f(p))$$

where $C(s)$ is the cluster assignment for tile $s$ and $d$ is a distance metric (we also use Euclidean distance for tile clustering). This clustering effectively projects the tissue sample onto the 1-dimensional path utilizing the learned spatial and histological similarity between tiles. With the cluster assignments,

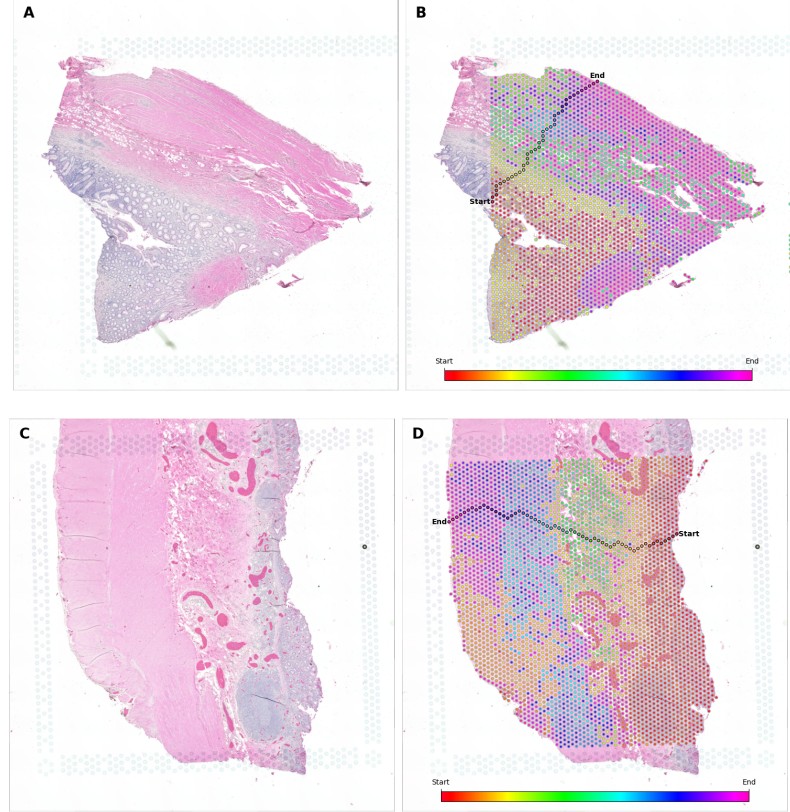

Figure 2: Traversal and alignment of colon tissue samples. (A) The H&E image of a colon sample from a patient with Crohn's disease. (B) The slide from A annotated by the hexagonal lattice grid. Between a user selected start and end point, the path traversed by ATAT is highlighted in black with each spot along the path colored by a rainbow gradient. The other spots on the grid share the same color as the spot along the path which is most visually similar. (C, D) The same as A and B for a patient with ulcerative colitis. The color gradient for B and D are distinctly mapped to their respective paths.

gene expression can be averaged within each cluster along the path $P^*$, resulting in 1-dimensional trajectories which can be further used for analysis (Fig 1C). The average gene expression along the path can be expressed as:

$$\hat{X}_{p,g} = \frac{\sum_{s \in S_p} \tilde{X}_{s,g}}{|S_p|}$$

where $\hat{X}_{p,g}$ is the average gene expression at tile $p \in P^*$ for gene $g$, $S_p = \{s|C(s) = p\}$ is the set of tiles $s$ assigned to cluster centroid $p$, and $\tilde{X}_{s,g}$ is the normalized gene expression of tile $s$ for gene $g$.

## 3   Results

We collect all 12 colon and 4 stomach samples together into a single tile dataset to train our encoder, as in Fig 1A. Here we present the path (Fig 1B) and aligned gene expression trajectories results for two colon tissue samples, one from a patient with CD (Fig 2A) and one from a patient with UC (Fig 2C).

Our method first aligns tiles across the whole slide to the most visually similar tile along the path, thereby clustering tiles from the same tissue layer together (Fig 1C); it accomplishes this by having learned the gradual visual transitions between tiles that are immediately adjacent to one another. As seen in Fig 2B,D, tiles from within the colonic mucosa are aligned to the tiles along the path

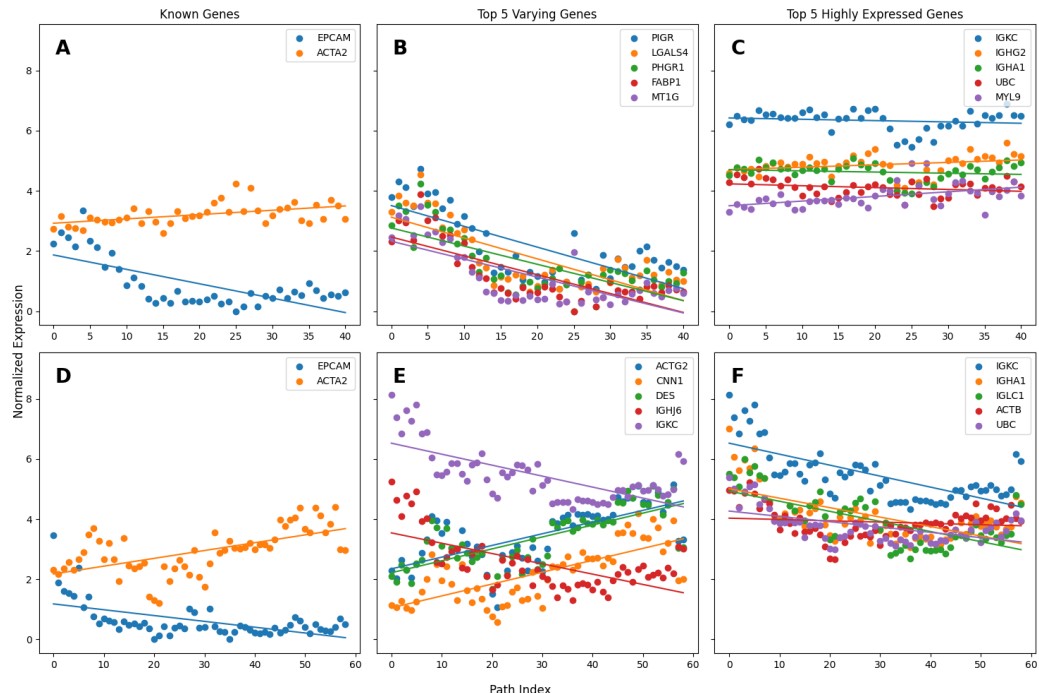

Figure 3: Gene expression trajectories along the traversed paths in Fig 2. (A-C) Expression of genes in tissue sample from patient with Crohn's disease, as in Fig 2A,B. (D-F) Expression of genes in tissue sample from patient with ulcerative colitis, as in Fig 2C,D. (A,D) Expression of known colon marker genes for intestinal epithelium, *EPCAM*, and smooth muscle, *ACTA2*. (B,E) Expression of top five genes ranked by standard deviation across the path, i.e. the most varying genes in each sample. (C,F) Expression of top five genes ranked by mean expression across the path, i.e. the most highly expressed genes in each sample.

which traverse the mucosa (spots colored in reds and yellows); other path segments similarly align their respective traversed layers to the tiles of the layers in the wider slide. However, there are some regions of spots which are not clustered according to their true layers; in the bottom-right section of Fig 2B, the muscularis mucosal region which should be colored in yellow-green are colored according to the purple-pink of the muscularis propria of the deeper tissue; in the bottom-left section of Fig 2D, the opposite is seen where the region of muscularis propria is colored in the yellow of the muscularis mucosa. We attribute this to the visual similarity between the microscopic morphology of the musuclaris layers, however we do acknowledge this potential limitation which warrants further investigation. Ultimately we recover gene expression trajectories which correspond to existing knowledge about colonic tissue layers and these diseases.

One additional feature of our method which we wish to highlight is the ability for ATAT to align non-linear tissue layers and complex morphologies to a path traversed through simpler, linear regions. As demonstrated in Fig 2A,B, despite our path only traversing the layers in the top portion of the tissue sample, the mucosal layers present in the bottom of the sample are also aligned by our algorithm to the traversed path. With the difficult reality of preparing neatly oriented tissue samples on slides and needing to potentially fit multiple tissue samples onto a single slide, our method allows for the joint analysis of all tissue pieces within the slide, regardless of morphology.

Once we have an aligned path, we plot known tissue and disease marker genes (Fig 1C). Fig 3A,D show the expression trajectories along the traversed paths of Fig 2B,D, respectively, for genes *EPCAM* and *ACTA2*. *EPCAM* is a known epithelial marker while *ACTA2* is expressed by smooth muscle. As our paths for both samples traverse from the colonic mucosa, through the submucosa, and end in the muscularis propria, we would expect *EPCAM* to decrease as the path index increases. We also observe an inflection point in the trajectory *EPCAM* in both Fig 3A,D at the approximate path

index ($\sim$ 10) which delineates the colonic mucosa from the submucosa. Inverse to *EPCAM*, *ACTA2* generally increases as tissue depth (path index) increases as smooth muscle becomes more abundant further from the colonic mucosa; likely due to the large submucosal layer in Fig 2D, a decrease in expression can be appreciated in the *ACTA2* trajectory in Fig 3D as the path traverses the submucosa.

We further rank the genes in both the CD and UC samples by the standard deviation (Fig 3B,E) and mean expression (Fig 3C,F) along the path. For the CD sample, the five genes with the largest standard deviation along the path (Fig 3B), i.e. the genes that change the most across tissue layers, correspond to epithelial gene markers [16]. In CD, marker genes relating to inflammation may not be highly variable along our traversed path, as CD is known for full-thickness inflammation of the tissue. We observe this constancy in the three genes with the largest mean along the path (Fig 3C), i.e. the genes which are most highly expressed; these genes (*IGKC*, *IGHG2*, and *IGHA1*) are related to plasma B-cell activity, a key mediator of inflammation in CD. In UC, a disease which typically only involves the colonic mucosa, we expect there to be similarity between the genes which are most highly expressed (and potentially related with disease) (Fig 3F) and the genes which are most highly variable across tissues (Fig 3E), as the inflammatory gene markers should be restricted to the depth of the mucosa. Indeed, *IGKC* is shared between the top five of either marker gene set and other similar genes involved in B-cell activity (*IGHA1*, *IGLC1*, *IGHJ6*) are included in either ranking. In all these genes, we observe a general decrease in signal with colonic tissue depth and an inflection point at the approximate path index which delineates the colonic mucosa.

## 4 Discussion

In our analysis of UC and CD samples, we observed a similarity concerning the immune cell type driving both diseases, while also identifying distinguishable variations in inflammation depth between them. We would like to mention that the standard deviation and mean expression of gene trajectories are relatively simple metrics for ranking and grouping genes. Consequently, integrating alternative signal processing metrics and incorporating complementary data modalities, such as single-cell RNA sequencing data, could potentially bolster this process. In this context, ATAT provides a basis to explore those additional data modalities as the trajectory alignment offers a biologically relevant reduced dimensionality for data exploration. Although the results for UC and CD do not introduce novel discoveries, they do confirm that our method consistently yields anticipated outcomes for diseases with recognized spatial phenotypes. Thus, ATAT presents a powerful machine learning method, enabling access to crucial molecular and cellular data in tissue biopsies and minimizing manual annotation in spatial transcriptomics analysis.

## 5 Acknowledgements

We thank the participants of our study whose data enabled us to develop ATAT. The authors thank Elizabeth Moison, Hugh Yeh, and Salvador Norton de Matos for insightful discussions. This work was partially supported by NIH DP2 NIAID New Innovator Award (DP2AI177884), the University of Chicago's Center for Interdisciplinary Study of Inflammatory Intestinal Disorders (C-IID) (P30 DK42086), the Institute for Translational Medicine (ITM) (5UL1TR002389-05), and a Hardware Grant from NVIDIA.

## 6 Code and data availability

The code for ATAT can be found at https://github.com/StevenSong/tissue-alignment. The repository includes code and documentation for training the vision encoder model and for path traversal and alignment. Data for colon and stomach tissue samples are available upon reasonable request to the corresponding author. Updated results are available on bioRxiv.

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
