# OpenReview forum: "ATAT: Automated Tissue Alignment and Traversal"
_NeurIPS.cc/2023/Workshop/AI4Science — NeurIPS2023-AI4Science Poster_

### Official Review · Reviewer_Jkic · 2023-10-24
**An automatic workflow for path finding of pathological data. More supporting evidence would help.**

**Rating:** 5
**Confidence:** 3

**Review:**

The authors proposed ATAT, an automated algorithm to infer a path across a sample with any morphology from H&E stained images. This path plays an important role when jointly analyzed with spatial transcriptomics, which simplifies the investigation of important biomedical questions.

This method changed the labor-intensive and expertise-required process into an AI-driven automatic process and therefore has great potential in real-world applications. The workflow is clearly described, and the machine learning model is trained and applied in a reasonable way. However, as a reader who hasn’t worked on pathological data before, I am not fully convinced by the manuscript that path finding + similarity searching can result in the best path.

Pros:

1 This method can be applied to any new samples without extensive re-training. It is fully automatic and can handle batch effects.

2 As stated by the authors, the gene expression trajectory from ATAT can be validated by prior pathological knowledge.

3 The manuscript is clearly written and easy to follow even for a researcher unfamiliar with pathological data.

Cons:

1 There are two steps that rely on “minimum distance” - Dijkstra path finding and tile alignment. However, the authors did not explain why the shortest path should be selected with concrete evidence. If d(a, b)=1, d(b, c)=5, d(c, d)=1, d(a, d)=8 and we are trying to find a path from a to d, with the authors’ strategy, a->b->c->d will be find. However, would a->d be another option (then b aligns to a, and c aligns to b)? Which one is better should be clarified with biological evidence.

2 I would also have been more convinced if some more experiments could be added in the result section. The authors only checked the standard deviation along the path, but the standard deviations for tiles at the same point on the path (i.e., the ones aligned together in Section 2.3.3) are not evaluated. Smaller standard deviations within these groups might better demonstrate the validity of the results.

3 I would recommend changing the color map of the path. The red color at both ends makes it harder to read the figures.

---

### Meta-Review · Area_Chair_P14n · 2023-10-27

**Recommendation:** Accept (Poster)
**Confidence:** 4

**Metareview:**

The paper is well-written and interesting, but it requires further technical explanation, more detail on experiments and results, and some minor revisions. There are concerns about using "minimum distance" without a clear explanation, and it's suggested to provide evidence to justify the choice of the shortest path. Additionally, including more experiments and evaluating standard deviations for tiles would strengthen the results.